# Overview of the Assays to Probe *O*-Linked β-*N*-Acetylglucosamine Transferase Binding and Activity

**DOI:** 10.3390/molecules26041037

**Published:** 2021-02-16

**Authors:** Cyril Balsollier, Roland J. Pieters, Marko Anderluh

**Affiliations:** 1Department of Chemical Biology & Drug Discovery, Utrecht Institute for Pharmaceutical Sciences, Utrecht University, P.O. Box 80082, NL-3508 TB Utrecht, The Netherlands; c.balsollier@uu.nl; 2Faculty of Pharmacy, University of Ljubljana, 1000 Ljubljana, Slovenia

**Keywords:** OGT, GlcNAcylation, *O*-GlcNAc Transferase, bioassay, OGT inhibitor

## Abstract

*O*-GlcNAcylation is a posttranslational modification that occurs at serine and threonine residues of protein substrates by the addition of *O*-linked β-d-*N*-acetylglucosamine (GlcNAc) moiety. Two enzymes are involved in this modification: *O*-GlcNac transferase (OGT), which attaches the GlcNAc residue to the protein substrate, and *O*-GlcNAcase (OGA), which removes it. This biological balance is important for many biological processes, such as protein expression, cell apoptosis, and regulation of enzyme activity. The extent of this modification has sparked interest in the medical community to explore OGA and OGT as therapeutic targets, particularly in degenerative diseases. While some OGA inhibitors are already in phase 1 clinical trials for the treatment of Alzheimer’s disease, OGT inhibitors still have a long way to go. Due to complex expression and instability, the discovery of potent OGT inhibitors is challenging. Over the years, the field has grappled with this problem, and scientists have developed a number of techniques and assays. In this review, we aim to highlight assays and techniques for OGT inhibitor discovery, evaluate their strength for the field, and give us direction for future bioassay methods.

## 1. Introduction

The study of *O*-GlcNAcylation, i.e., the reversible enzymatic posttranslational attachment/detachment of the *N*-acetylglucosamine sugar moiety to proteins, is a challenging topic in biochemistry and medicinal chemistry. Interestingly, this modification is regulated by only one pair of enzymes: *O*-GlcNAc transferase (OGT), which catalyses the post-translational transfer of *N*-acetylglucosamine via *O*-glycosidic linkage to the serine and threonine residues of proteins, and *O*-GlcNAcase (OGA), which can remove them (Figure 1) [1,2,3]. This topic has received a major boost in the last two decades with the discovery of small molecule inhibitors that modulate the process. Many studies have revealed the ubiquitous role of *O*-GlcNAcylation in biological processes, such as gene expression [3,4], cell cycle control [4,5], and regulation of enzyme activity [6,7,8,9]. Moreover, OGT competes with kinases for the modification site, leading to competition and cross-regulation of these translational modifications [9,10,11]. Dysfunction in the O-GlcNAcylation pathway has been associated with various chronic diseases [9,12,13]. In diabetes, it is associated with cognitive complications [12], while aberrant *O*-GlcNAcylation of tau protein in the brain leads to possible early onset of Alzheimer’s disease [13]. In cancer, *O*-GlcNAcylation appears to regulate tumour spread through adhesion and proliferation of cancer cells [1,14,15].

To study the mechanism of *O*-GlcNAcylation and the consequences of *O*-GlcNAcylated proteins in these diseases, molecular tools are needed to control the level of *O*-GlcNAcylation, i.e., inhibitors of both enzymes that modulate the *O*-GlcNAcylation status of proteins. In the case of OGA, many potent inhibitors have already been developed to control tauopathy levels in the brain. Two of them are currently in clinical trials for the treatment of Alzheimer’s disease and progressive supranuclear palsy (PSP), two diseases associated with tauopathy levels: MK-8719, which has passed preclinical trial tests, and ASN120290, which is scheduled for phase 2 clinical trials [16,17,18]. In contrast, the development of OGT inhibitors is less advanced and still in its infancy. As a transferase, OGT binds both the cofactor UDP-GlcNAc and a substrate protein. This provides additional opportunities, as blocking either of the two substrates should inhibit the enzyme. The goal of blocking both substrate and cofactor binding sites would likely result in inhibitors with molecular mass and physicochemical properties that would make these inhibitors non-drug-like. Moreover, as evident from some studies, the lack of stability of the isolated enzyme is a problem that complicates the search for potent inhibitors. It makes protein expression and purification a tedious practice, making in vitro assays with OGT problematic even on a routine basis [19].

Several strategies for designing OGT inhibitors have been described in the literature. We and others have reported bisubstrate/conjugate inhibitors consisting of a UDP mimic and a peptide fragment such as Goblin 1 or F20 [20,21]. OGT inhibitors based on UDP-GlcNAc mimics, such as UDP-5S-GlcNAc [22], have also been reported. Next, there are the UDP-mimics of the OSMI family, with OSMI-4 as the most potent binder to date with a K_d_ of 8 nM (Figure 2) [23,24,25]. To test new inhibitors, specific tools are needed, especially bioassays that can give a rapid and direct indication of the OGT inhibitory activity and the binding properties of the OGT inhibitor. Nowadays, with the development of new optical methods and the wider availability of high-throughput screening methods, there are many ways to design bioassays tailored to a specific macromolecular target. Somehow, this does not seem to be the case for the development of assays to assess the enzymatic activity of OGT and/or inhibitory potency of OGT inhibitors. The first reason is the instability of OGT [19]. As a result, despite many attempts, none of the published assays became a reference standard for the field. In this review, we provide an overview of bioassays reported in the recent literature for testing OGT activity. We also highlight the most suitable assays and point out the new bioassays for screening OGT inhibitors (Figure 3). The first part of this review covers assays for measuring enzyme inhibition, kinetics, and different ways to follow the enzymatic response. The final part discusses assays that measure the binding affinity of the OGT inhibitors.

## 2. Assays for Evaluation of GlcNAc-Transferase Activity

O-GlcNAc transferase uses UDP-GlcNAc as a donor substrate to transfer the GlcNAc moiety to selected serines and threonines on protein substrates, resulting in GlcNAcylated proteins and UDP as the reaction products. An assay to monitor OGT enzyme activity may rely on measurement of the two resulting products, which must be quantified. Interestingly, UDP as a by-product is a rather potent OGT inhibitor with an IC_50_ of 1.8 μM [26]. This must be taken into account when using a bioassay to test inhibitory potency, as an inhibitor is formed as the reaction proceeds. In this first part, we discuss such assays and how they have advanced the field.

### 2.1. UDP-Glo Assay

The UDP-Glo assay developed by Promega is a tool for studying glycosyltransferases in general, as it can be used for any UDP sugar donor [27,28]. It measures the UDP formed during a glycosyl transfer reaction. It is a paired assay that involves several enzyme reactions. First, the O-GlcNAcylation reaction occurs by introducing UDP-GlcNAc as a donor, along with a peptide substrate and OGT. The second reaction step is catalysed by a cytidine monophosphate kinase (CMK), which transfers a phosphate from UDP to ADP present in the detection solution. The ATP formed is further processed with oxygen by a luciferase enzyme that oxidises luciferin to oxyluciferin, AMP, and CO_2_ that emits luminescence (Figure 4) [29]. The measured luminescence is proportional to the UDP concentration produced during the reaction and thus indirectly reflects the extent of GlcNAcylation.

This assay can detect very low concentrations of UDP due to signal amplification from the luminescence reaction. Detection is possible from µM down to the low nM range [29]. The amplification effect of the coupled reaction makes this assay highly sensitive. In this way, amounts of unstable substances can be reduced, such as the concentration of OGT or other reagents. The main disadvantage of this method is the indirect measurement due to the three coupled reactions. For example, UDP-GlcNAc may spontaneously hydrolyse to UDP in aqueous solution, resulting in a higher final value and false negative results, i.e., low apparent inhibition. In contrast, the coupled enzymatic reaction that detects UDP can be inhibited by UDP mimics, leading to false positives.

The UDP-Glo assay has been used to screen and evaluate different types of inhibitors. Peptide inhibitors derived from protein substrates of OGT such as RBL-2 and ZO-3 were evaluated and gave IC_50_ values of 385 µM and 184 µM, respectively [21]. Bisubstrate inhibitors of these peptides were also prepared, resulting in improved potency with an IC_50_ value of 117 µM for the most potent compound [21]. Compound L01, discovered by screening natural products, was also evaluated with UDP-Glo, resulting in an IC_50_ of 22 µM [30]. Recently, several peptidomimetics were also screened and evaluated with UDPGlo, resulting in LQMed 330 and LQMed 269 with IC_50_ of 11.7 and 159 µM, respectively (Figure 5) [31].

### 2.2. [14. C] UDP-GlcNAc

In 1978, Mendicino et al. synthesised a radioactive UDP-[1-^14^C]-GlcNAc from [1-^14^C]acetate [32], which was further used to measure OGT activity by Hanover et al. in 1995. The idea was to perform the O-GlcNAcylation reaction with ^14^C-labeled UDP-GlcNAc, which allowed quantification of the GlcNAcylated target protein. After completion of the reaction, the proteins are separated in an SDS-page gel and incorporation of [^14^C]-GlcNAc is assessed by autoradiography of the gel. The kinetics and viability of the substrate tested can then be correlated with the half-life of [^14^C]. The assay allows for a very sensitive final evaluation but uses radioactive materials. This makes it expensive and requires special biosafety measures, so it is less suitable as a primary screening method for OGT inhibitors.

In 2005, the Walker group used this method to confirm hits from other assays and obtain an IC_50_ measurement for the OSMI family [23,24,33]. The compound UDP-5S-GlcNAc was also screened using this radiolabelled assay and gave an IC_50_ of 8 µM (Figure 6) [34].

### 2.3. FRET Assay

In 2008, Walker reported a novel OGT inhibition assay [29] based on protein resistance to proteolysis enhanced by O-GlcNAcylation. The assay requires a specific peptide—a dual substrate for both OGT and a protease. The peptide is conjugated at one end to a fluorescent donor and at the other end to a fluorescent acceptor to allow fluorescence resonance energy transfer [30] (FRET). The fluorescent pair is carefully selected so that the emission wavelengths of the donor molecule and the excitation wavelengths of the acceptor molecule overlap. FRET requires a short distance between the two fluorophores and resonance occurs most efficiently when they are within 10 nm.

In the case of the FRET assay with OGT, cleavage of the peptide substrate is inhibited when the peptide substrate is GlcNAcylated (Figure 7). In the presence of an OGT inhibitor, glycosylation of the protein is stopped and the decrease in FRET signal can be detected. The subsequent proteolysis is proportional to the inhibitory activity of an OGT inhibitor. The assay gives a clear indication of the inhibitory potency of the compound tested, as not only the concentration of the two molecules can be measured but also the kinetics of the reaction. This method monitors the presence of glycosylated peptide and can be considered a direct method. However, quantification of the degree of glycosylation by the protease occurs in a second step, which can be problematic when screening large libraries that require rapid measurement and minimal steps. Furthermore, the apparent specificity of the protease for the non-GlcNAcylated peptide is usually due to higher reaction kinetics. In fact, the GlcNAcylated peptide can be cleaved, but with slower reaction kinetics. This makes it difficult to set up and run the assay in a high-throughput screening format (HTS), as running the assay in multi-well plates results in a delay in reagent addition and/or fluorescence readout.

According to the authors of the paper [29], new molecules were found using this assay (Figure 8) that were evaluated using the radiolabelled assay, but no data were presented regarding binding or inhibition for specific molecules. It is only mentioned that the compounds had an IC_50_ between 0.9 and 20 µM.

### 2.4. PamStation

The PamStation is a product developed by PamGene Company (PamGene International B.V.). This method was developed to discover novel kinase substrates/inhibitors for cancer therapy and prognostic purposes. Since O-GlcNAcylation and phosphorylation can occur at the identical modification sites (e.g., the amino acid residues Ser and Thr) or at nearby residues, PamStation was used to identify OGT substrates and then to study the crosstalk between the two translations [21,35,36,37]. This method is based on specific innovations. The first is a special 4- to 96-well microplate called the PamChip, in which each well contains a 3D porous alumina support. Within these wells is a high-density surface coating of various peptides that transforms the well into a dense microarray [38]. The chips are placed in the PamStation apparatus, a machine that pumps the reaction solutions from the bottom of the porous material using an innovative tide-like flushing technique. An image of the well is taken at regular intervals to collect data. During this process, the fluorescent liquid is temporarily below the chip and out of range of the camera. The fluorescence intensity of each well is measured, indicating whether modification of the peptides on the chip surface has occurred. The station allows programming of a high number of cycles to perform kinetics measurements in real time (Figure 9). The enzymatic reaction consisted of OGT and UDP-GlcNAc loaded onto a PamChip. When the reaction occurs, the GlcNAc moiety is transferred to the peptide, allowing the primary antibody to bind to GlcNAc, while final detection is achieved with a FITC-labelled secondary antibody.

The PamStation is a robust tool for evaluating peptide modifications and has been successfully used to screen OGT substrates that were transformed into inhibitors after mutation and linked to a quinolone scaffold to form bisubstrate inhibitors (Figure 10) [21,35,36,37]. The disadvantage of this technique is the use of a pair of antibodies by which an indirect measurement of glycosylation is made. Due to occasional nonspecific binding of the antibody, signal enhancement may be detected without glycosylation, leading to false positive results. Furthermore, the PamStation and PamChip are not standard equipment in many laboratories, but this should not prevent any scientist from using and further developing this efficient technique.

### 2.5. Measurement of OGT Activity by HPLC

In general, LC-MS can be used to monitor glycosyltransferase activity. The method is based on the difference in retention time and molecular weight between a peptide and its glycosylated derivative. This difference allows their qualitative separation and quantification. Zhang took the CKII peptide, a known OGT substrate, and incubated it with OGT and UDP-GlcNAc. After O-GlcNAcylation, the sample is spun down and the supernatant is separated on a C18 analytical column. The relative comparison of the glycopeptide peak integrals compared to the native peptide peak gives us information about the enzyme activity. By preparing a gradient concentration of a potential inhibitor, the method can be used to determine the corresponding IC_50_. This method can be easily used in many laboratories as the HPLC-MS coupled spectrometer is standard laboratory equipment. The main disadvantage of the method is that the reaction mixture must be purified before injection. This step makes the assay unsuitable for HTS, so it is recommended to use an orthogonal assay for hit screening. The screening of natural products that led to the discovery of the L01 inhibitor was performed using this method and then confirmed by using the UDP-Glo assay [30].

### 2.6. Fluorescence Intensity Assay

The Vocadlo group published a new bioassay for screening OGT inhibitors in HTS based on a chemically modified UDP-GlcNAc [39]. The modified GlcNAc was prepared by extending the amino group of the corresponding glucosamine precursor with a Bodipy fluorescent group (Figure 11).

This UDP-GlcNAc Bodipy cofactor is used by OGT as a labelled glycosyl donor to transfer the fluorescent GlcNAc to an acceptor peptide. A modified CH1 peptide, a known OGT substrate, is used as an acceptor and is conjugated to biotin. After performing this enzymatic reaction, the GlcNAcylated peptide is captured with a streptavidin-coated device, allowing easy washing to remove the excess fluorescent material. The fluorescent signal is measured to allow direct quantification of glycosylation [40] (Figure 12).

This assay was used to perform a high-throughput screening on a library of known bioactive compounds [39]. The screening led to the discovery of Dingo-4a as a potent OGT inhibitor with IC_50_ of 6 µM (Figure 13), which was originally developed as a target for dynamic GTPase (Dynl).

Further studies on the scaffold showed that it can be made OGT specific and could then be a starting point for a new family of inhibitors.

## 3. Biophysical Assays to Measure Binding Affinity of a Potential Inhibitor

Another way to screen OGT inhibitors is to use a biophysical method to measure the binding affinity of a ligand for its target. This can vary in different experimental setups, but still relies on binding phenomena at the binding site of the macromolecule, which is less susceptible to experimental change. Most biophysical techniques are straightforward and are suitable for screening purposes. Since inhibitors’ activity can be quantified by IC_50_ values that are not directly comparable to binding constants, biophysical assays should be used as orthogonal assays to confirm inhibitors’ binding characteristics.

### 3.1. Fluorescence Polarization Displacement Assay

The fluorescence polarization (FP) technique is a reliable and rapid screening method for assessing the affinity of small molecules competing with a fluorescently labelled probe of known K_d_ value [41]. This method is based on the difference in light polarization between a free fluorescent probe in solution and a bound probe. Low polarization is due to the very rapid molecular rotation and movement of a small molecule, while high polarization occurs when the probe is bound to a larger molecule. This difference can be seen when a probe is irradiated with polarised light.

A FP-based displacement assay requires a fluorescent probe that binds strongly to the protein of interest. The fluorescent probe will form a complex in the presence of the protein, giving high polarization. When a competitive ligand is added to the solution, it outcompetes the fluorescent probe, and its binding potency correlates with the decrease of the signal. If the polarization does not change, the ligand does not compete with the probe and is therefore not a binder (Figure 14). The major drawback of the method is that only competitive binders can be detected. Allosteric binders would not show up as hits. Two FP probes for OGT have been developed, one by the Walker group [24] and the other by the van Aalten group [42].

#### 3.1.1. Walker Group FP Assay

As a probe, the Walker group chose to synthesise a UDP-GlcNAc/fluorescein derivative by extending the acetyl-amine group with a linker (Figure 15). The probe affinity for OGT was measured to be 1.3 µM.

To see if the probe meets the requirements of a displacement assay, a test must be performed with a known binder to correlate the accuracy of the assay and the ability of the probe to be displaced. A gold standard in practice is to use the original binder from which the probe was developed. In this case, UDP and UDP-GlcNAc were used as validation. This assay led to the discovery of quinolone-6-sulfonamide as potential scaffolds. Optimization of these quinolone-6-sulfonamide hits led to the development of the OSMI family (Figure 16) [24,25].

#### 3.1.2. Van Aalten Group FP Assay

The probe used by the van Aalten group for the FP assay originates from their bisubstrate inhibitors. It consists of the UDP-part introduced as an Allyl-UDP, which reacts with the cysteine of a peptide conjugated to fluorescein moiety (Figure 17) [42]. The displacement of the fluorescent probe was tested against similar compounds, including UDP-5S-GlcNAc and two bisubstrate inhibitors. After displacement, the K_i_ found for UDP-5S-GlcNAc was consistent with data from other assays, validating the method. This assay was used to examine the inhibitory potency of various peptide sequences bound to S-linked UDP.

#### 3.1.3. Conclusions

In summary, fluorescence polarization is the first assay used in the OGT field that identified nM hits. It does not require secondary steps prior to readout. The main disadvantage of the FP method is that it requires a specific probe that is complicated to design. Many parameters can affect the polarization properties, such as hydrophobicity, linker length, or the attachment point of the fluorophore. For screening purposes, it is most important that the probe has a high affinity for the target, since its K_d_ value determines the detection limit of the displacement assay. Apart from this, FP assays can be considered very useful for primary screening. In the OGT field, the latest OGT inhibitors show binding affinity of the same or higher order of magnitude as the two probes presented previously. To find even more potent inhibitors, there is a need for more efficient probes to increase the sensitivity of the FP assay in the hit selection process

### 3.2. Microscale Thermophoresis

Microscale thermophoresis (MST) is an optical technique based on two physical phenomena: temperature-related intensity change (TRIC) and thermophoresis [43]. TRIC describes a temperature gradient effect on the fluorescent molecule. Namely, changing the temperature of a solution changes the fluorescence properties of the fluorophore, modifying the fluorescence intensity. Thermophoresis is a phenomenon that describes how fast molecules move under the influence of an increasing temperature. In MST, these two effects are used in synergy to measure or monitor the movement of the ligand–protein complex under the influence of a changing temperature. During an MST experiment, the solution containing the labelled protein and the non-fluorescent ligand is irradiated with a combination of an infrared laser and light at the wavelength of the probe excitation. This combination of light causes a decrease in the measured fluorescence intensity, until a plateau is reached that corresponds to the thermophoresis motion of the probe. This plateau varies depending on the microenvironment around the probe and is therefore affected by the concentration of the ligand. By observing the thermophoresis of the probe at different ligand concentrations, a binding curve can be drawn indicating the K_d_ value of the ligand against a selected target. This technique has several advantages: first, the labelled element is the protein, avoiding the problem of using a small molecule as a probe. Additionally, by labelling the protein, the cell lysate can be measured directly, eliminating the need for protein purification, since other proteins present in the mixture would not give a signal. MST is a powerful technique, but it requires a special reader. In addition, the containers for the solution are disposable capillaries, which are still not standard equipment in a laboratory. Regarding OGT, the Walker group published a method for measuring binding to pure OGT for the OSMI compounds including OSMI-4 [25].

### 3.3. Tethering In Situ Click Chemistry

Tethering in situ click chemistry (TISCC) developed by Zhang et al. is a method that can be classified as a target-guided synthesis [44]. The goal is to perform a click reaction within the binding site between libraries of chemically reactive fragments. The protein binding site thus serves as a “mould” for the design of a new inhibitor. These new hits, composed of fragments, are resynthesised and tested with an orthogonal assay to assess their potency against the target. The reported method uses two different chemical libraries [44]. The first library consists of molecules with an iodoacetamide moiety as well as an adjacent azide group. The iodoacetamide moiety is a tethering group that allows the molecule to form a covalent bond with any cysteine on the protein surface. In the first step, OGT is added to a solution containing a starting library so that the tethering group can react with the cysteine that is in contact with the active site. Then, the newly attached moieties with free azido groups perform thermal/non-catalysed Huisgen cycloadditions. Subsequently, the second library containing the binding fragments bearing an alkyne moiety is introduced under click chemistry reaction conditions. The protein itself serves as a mould and harbours only fragments with reasonably high affinity for the binding site so that a click reaction occurs only with the high-affinity alkyne fragments. After the click reaction, the protein is separated from the molecules with SDS page and digested with trypsin. The molecules are then analysed by LC/MS/MS to identify fragments that participated in the click reaction. The complete molecules are then resynthesised, and their inhibitory potency is measured using the commercially available UDP-Glo assay. Through the selection process, Zhang et al. identified two inhibitors APNT and APBT with a measured IC_50_ of 68 µM and 169 µM, respectively (Figure 18). After further testing, these compounds were found to be cell-permeable inhibitors and non-competitive for UDP-GlcNAc.

The main disadvantage of this method is that it targets both the binding site and any other pocket large enough to contain one of the other fragments. On the other hand, this methodology can help to identify allosteric pockets and possibly find new chemotypes and modes of OGT or other inhibition. Another disadvantage of this methodology is that the number of hits is usually very high. Since this method cannot discriminate between orthosteric and allosteric inhibitors and all compounds must be re-synthesised to be re-tested on an orthogonal assay, the cost is very high.

## 4. Discussion

The diversity of methods and the extensive development of this field show how challenging the study of OGT is. It also shows the evolution of GlcNAcylation studies, from the need to see and evaluate GlcNAcylation to the search for selective and potent inhibitors to validate OGT as a therapeutic target. To achieve this goal, the field needs a simple assay that can both screen large libraries and evaluate the enzyme inhibitory activity of hits.

As described above, enzyme kinetics and inhibitory activity can be measured using two types of assays. The first category uses an indirect approach by measuring the concentration of UDPas GlcNAcylation products or by detecting GlcNacylation on a peptide/protein substrate. The other category uses a direct approach, either by using antibodies to visualise the extent of GlcNAcylation with a PamStation or by using a modified cofactor bearing a fluorescent label, such as the Vocadlo group’s fluorescence intensity assay. Among the indirect approaches, there are several methods, such as the UDP-Glo and the FRET assay. These techniques use enzymatic reactions to modify the products of the original reaction to monitor GlcNAcylation, and have the advantage of not affecting the OGT reaction itself. On the other hand, the additional steps add other factors (inhibition of the enzyme, non-selective binding, kinetics) that can interfere and change the result, leading to false negatives or false positives. Other commercial assays can be used on the OGT: a) the Transcreener Assay from Bellbrooks lab, which uses a UDP-specific antibody linked to a fluorophore to monitor UDP production by fluorescence polarization; and the Malachite Green Assay from R&D Systems, which uses phosphatase to form pyrophosphate from the UDP form and then allows this pyrophosphate to react with malachite, forming a green complex that can be quantified by luminescence. We have not reported on these assays due to the lack of literature data on screening of OGT inhibitors.

The direct approach in assays is used to directly measure glycosylation of the acceptor substrate. The examples given here are the HPLC/MS method, a method using a radioactive donor substrate, and the fluorescence intensity assay. Of these three, two of them have the advantage of not interfering with the OGT reaction because the HPLC/MS measures GlcNAcylation by identifying the molar mass of the GlcNAcylated product. Moreover, the enzymatic activity is not disturbed by the change of a carbon to its radioactive isotope. The fluorescence intensity assay, however, with the linkage of the donor substrate to the Bodipy fluorophore, alters the kinetics of OGT. This problem was solved by the Vocadlo group with an extensive kinetic study of the fluorescent cofactor(s). The fluorescence intensity assay developed by Vocadlo meets most of the requirements when reliability is considered. The method is easy to use, requires no additional steps prior to readout, and allows screening for inhibitors in an HTS manner. This assay is not commercially available and therefore requires some preparation. In contrast, the UDP-Glo is available as a kit and can be easily used as an HTS assay for various glycosyltransferases.

Three biophysical methods have been used to quantify OGT binding events. The first is microscale thermophoresis, a method that has been used to calculate the K_d_ of the OSMI family. Importantly, this was the only reported method sensitive enough to provide binding constants for inhibitors in the nanomolar range such as OSMI-4. It also eliminates many problems (purification, activity, expression yield, etc.) associated with OGT. The main drawback of this method is that it is not suitable for HTS and should rather be used as a confirmatory method for binding constants. The second biophysical method, called tethering in situ click chemistry, is an in vitro chemical method that allows processing of a large number of fragments where the protein itself picks the right binders. However, it targets the entire protein surface, which can result in molecules that bind allosterically to the protein surface or have no effect on enzyme activity. Moreover, this method requires resynthesis without the tethering group and reassessing the binding affinity with an orthogonal assay. The third method is fluorescence polarization, a technique commonly used in medicinal chemistry because of its rapidity in screening setups. Furthermore, the method does not require manipulation of the target protein; the only requirement is a molecular probe or ligand with a fluorophore attached. The final readout is the measurement of fluorescence polarization, the intensity of which corresponds to the displacement of the fluorescent probe. The tricky part of the method in these assays is the design and synthesis of a fluorescent probe that is a strong binder and allows the measurement of FP. Two fluorescent probes have been synthesised for OGT, and one has been used successfully by the Walker group to perform HTS. This technique is probably the most suitable to screen large compound libraries. The only problem found is that the current inhibitors have leapfrogged the affinity of the actual fluorescent probe. If a more potent fluorescent probe is developed in the future, this could be the key to improved OGT inhibitor discovery. All methods discussed here are summarised and presented in the final table (Table 1).

Cellular assays and cellular uptake of inhibitors were not discussed here. One of the few cellular assays for testing OGT inhibition is a Western blot using anti-O-GlcNAc antibodies to measure the total glycosylation inside the cell. The disadvantage of this method is that it is not known whether the observed inhibition is from OGT inhibition of the compound tested or from an off-target cytotoxic effect. Another such assay reported by Hu et al. [45] uses the unnatural metabolic probe UDP-6AzGlcNAc with an azide handle, which can be readily detected using click chemistry (attachment of a fluorescent probe with alkyne functionality) and in-gel fluorescence scanning. New types of cellular assays are needed and will be developed in the coming years that will provide useful information to make OGT a valid therapeutic target.

## Figures and Tables

**Figure 1 molecules-26-01037-f001:**
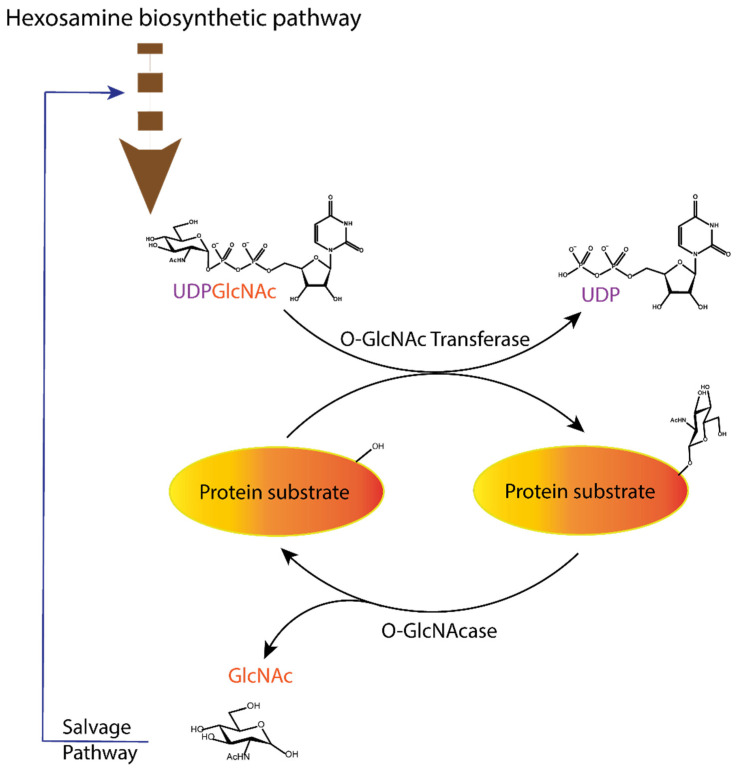
Dynamic equilibrium of the O-GlcNAcylation process. UDP-GlcNAc with the sugar moiety attached via α-*O*-glycosidic bond to the UDP is produced in the hexosamine biosynthetic pathway and subsequently attached to serine/threonine of protein substrates by *O*-GlcNAc transferase (OGT) via *O*-β-glycosidic bond. It is removed by *O*-GlcNAcase (OGA), and the GlcNAc formed after the OGA reaction re-enters the hexosamine pathway.

**Figure 2 molecules-26-01037-f002:**
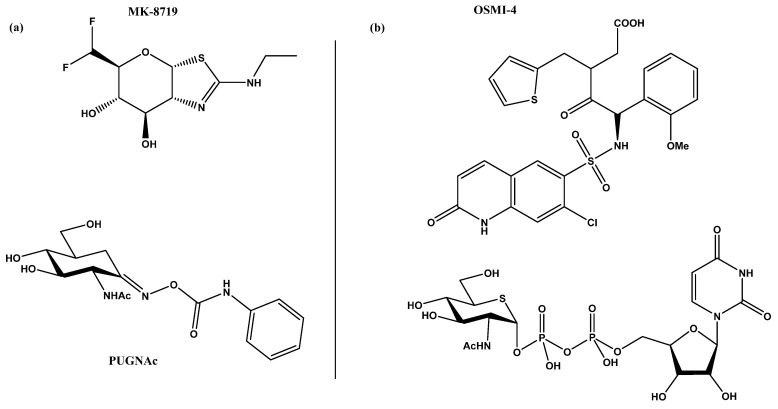
(**a**) MK-871914 and PUGNAc20 OGA inhibitors; MK-8719 potent clinical candidate as OGA inhibitor; (**b**) OSMI-4, the most potent OGT inhibitor to date, and UDP-5S-GlcNAc, an inhibitor biosynthesised by hijacking the hexosamine pathway using a per-acetylated 5S-GlcNAc.

**Figure 3 molecules-26-01037-f003:**
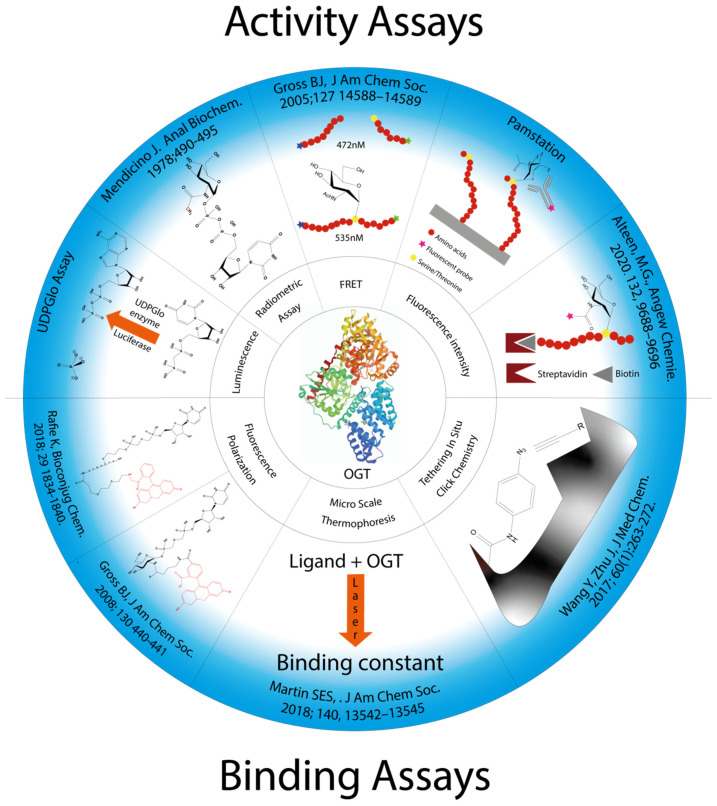
Presentation of all methods reported in the literature for screening OGT inhibitors and the techniques on which they are based.

**Figure 4 molecules-26-01037-f004:**
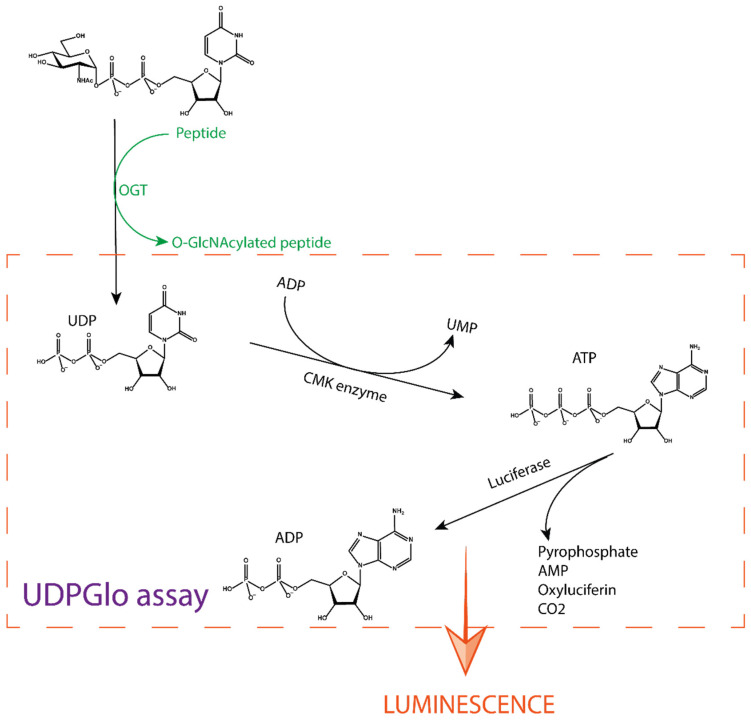
Schematic approach of the UDP-Glo assay. After the production of UDP by the OGT reaction, a specific buffer is added to stop the reaction. This buffer also contains ADP and the cytidine monophosphate kinase (CMK) enzyme that converts UDP to UMP and ADP to ATP. The ATP formed reacts with oxyluciferin in a luciferase-catalysed reaction. This reaction yields carbon dioxide, AMP, pyrophosphate, and luminescence. The amount of light produced is directly proportional to the concentration of UDP formed in the OGT reaction.

**Figure 5 molecules-26-01037-f005:**
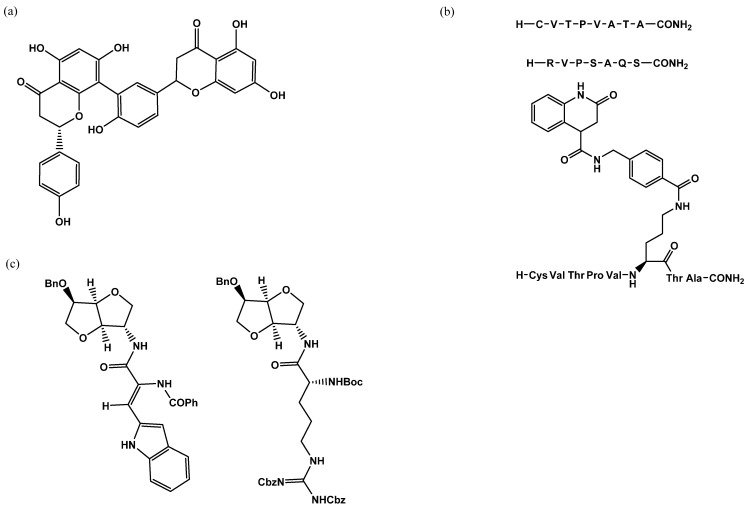
OGT inhibitors assessed by UDP-Glo assay: (**a**) L01 IC_50_ = 22 µM; (**b**) Pep6 IC_50_ = 385 µM, Pep13 IC_5O_ = 193, and bisubstrate inhibitor compound **6** IC_50_ = 117 µM; (**c**) LQMED 269 IC_50_ = 159 µM and LQMED 330 IC_50_ = 11,7 µM.

**Figure 6 molecules-26-01037-f006:**
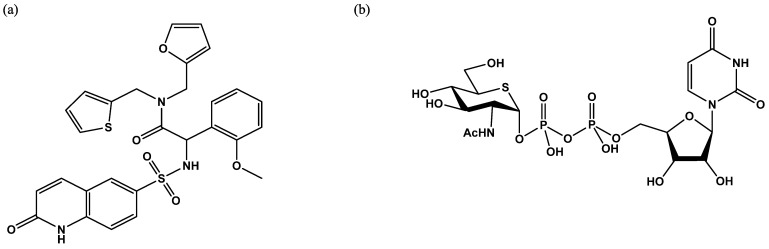
Inhibitors assessed with the radiolabelled assay: (**a**) OSMI-1 IC50 = 2.7 µM (**b**) UDP-5S-GlcNAc IC50 = 8 µM.

**Figure 7 molecules-26-01037-f007:**
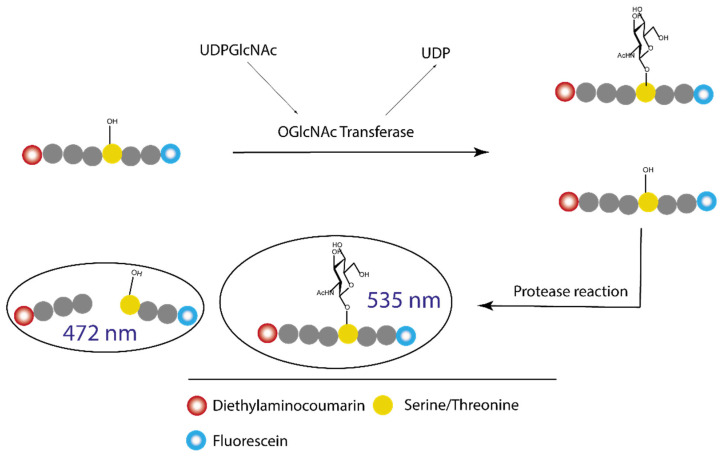
Schematic representation of the fluorescence resonance energy transfer (FRET) assay developed by the Walker group [29]. The GlcNAc acceptor is loaded with two fluorescent probes, diethylaminocoumarin and fluorescein. After the OGT reaction, a portion of the peptide is OGlcNAcylated. The mixture of GlcNAcylated and non-GlcNAcylated peptides is then mixed with a protease that cleaves only the free peptide, preventing FRET.

**Figure 8 molecules-26-01037-f008:**
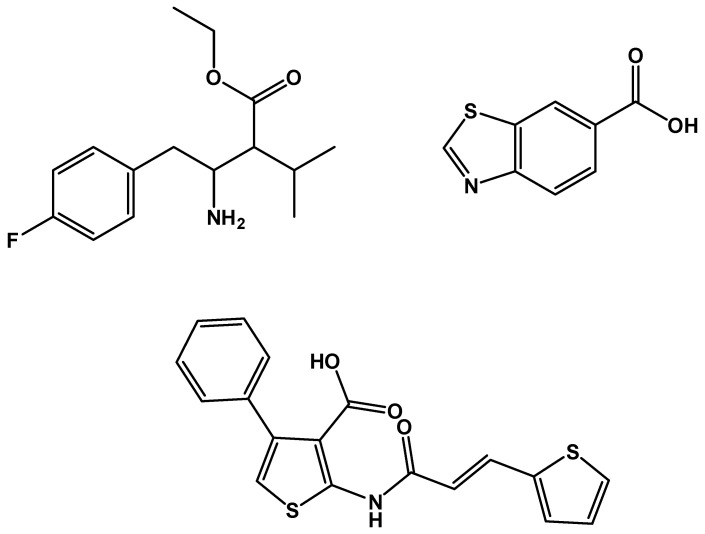
Some of the inhibitors uncovered by the FRET assay performed by Walker et al. [29].

**Figure 9 molecules-26-01037-f009:**
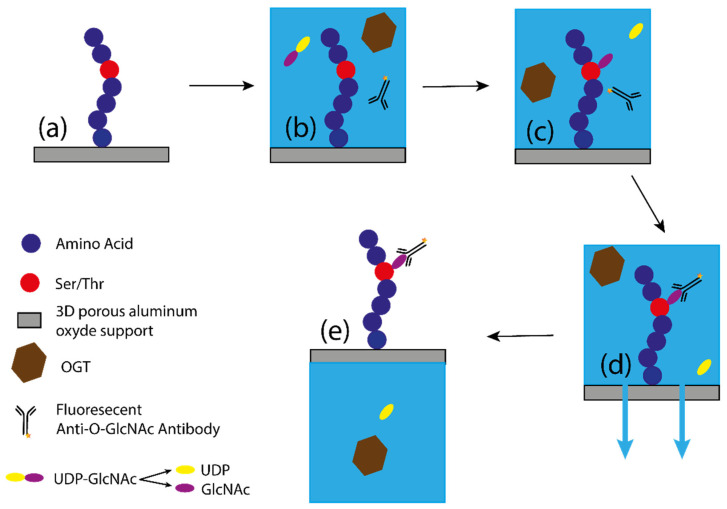
Schematic representation of O-GlcNAcylation by the PamStation [33]. (**a**) Representation of the peptide bound to the 3D porous alumina support; (**b**) the solution containing OGT, substrates, and antibody is added to the porous material; (**c**) the peptide is O-GlcNAcylated, releasing UDP in solution; (**d**) after the glycosyltransferase reaction, the antibody binds to the GlcNAc. The buffer is then pumped down through the porous material with only the fluorescent antibody bound to the GlcNAc, allowing measurement of fluorescence intensity. (**e**) The buffer is then pumped back with the peptide to complete one cycle. A measurement of fluorescence intensity is taken at each cycle, allowing the kinetics of the reaction to be measured.

**Figure 10 molecules-26-01037-f010:**
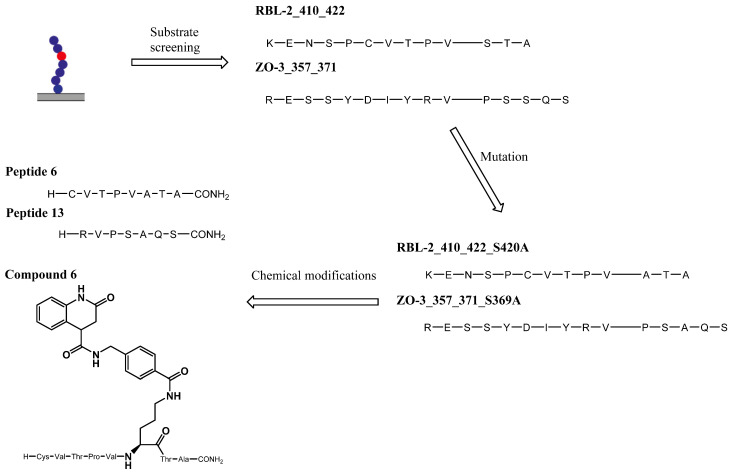
Development of peptidic and hybrid inhibtors of OGT starting with PamStation substrate screening.

**Figure 11 molecules-26-01037-f011:**
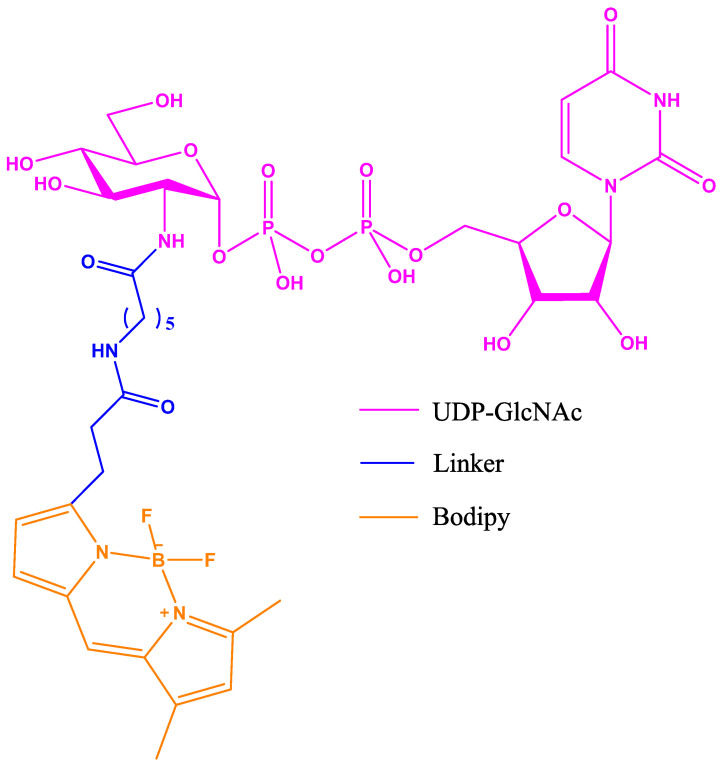
UDP-GlcNAc with Bodipy fluorescent probe attached via a modified acetyl group at position 2 of the UDP-GlcNAc.

**Figure 12 molecules-26-01037-f012:**
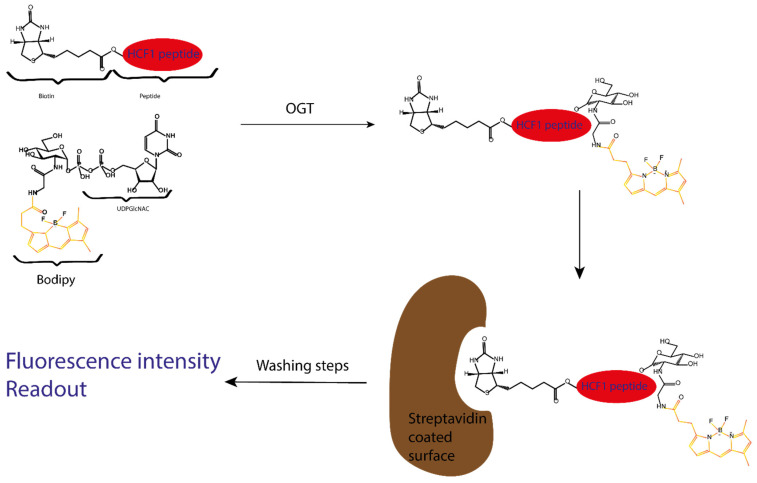
Schematic representation of the Vocadlo group fluorescence intensity assay. First, the biotinylated peptide acceptor is GlcNAcylated in the presence of OGT and the fluorescently labelled UDPGlcNAc. After the reaction, a mixture of fluorescent GlcNAcylated peptide and the native biotinylated peptide is present: the ratio of both reflects the reaction rate, which diminishes if OGT is inhibited.

**Figure 13 molecules-26-01037-f013:**
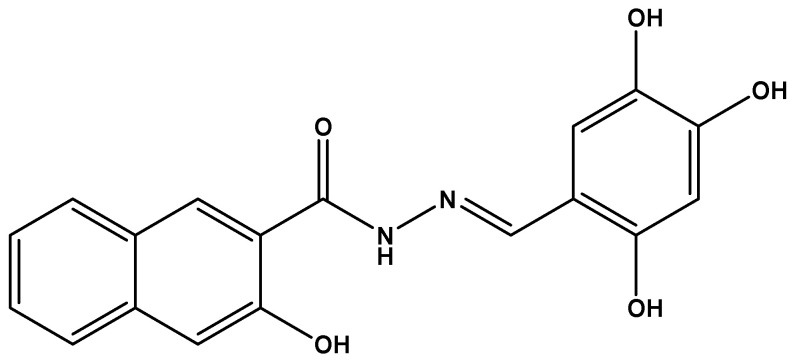
Structure of Dyngo-4a, potency against OGT: IC_50_ = 6 µM; potency against dynamic GTPase (Dynl): IC_50_ = 2.7 µM.

**Figure 14 molecules-26-01037-f014:**
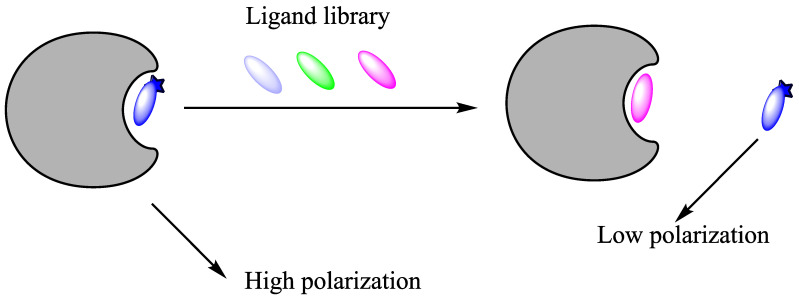
A schematic presentation of a fluorescence polarization displacement assay. First, the enzyme and the fluorescent probe are mixed in a solution: the probe binds to the active site of protein, forming a complex that will give a high polarization; competitive inhibitor/ligand will displace the probe and release it into the bulk phase that will result in decrease of light polarisation.

**Figure 15 molecules-26-01037-f015:**
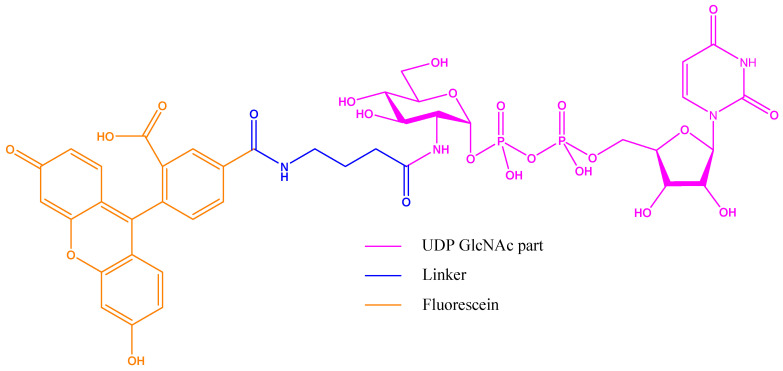
Fluorescent probe designed by Walker group in their fluorescence polarization (FP) assay: the binding part is UDP-GlcNAc with the extended acetyl group as linker to a fluorescein derivative.

**Figure 16 molecules-26-01037-f016:**
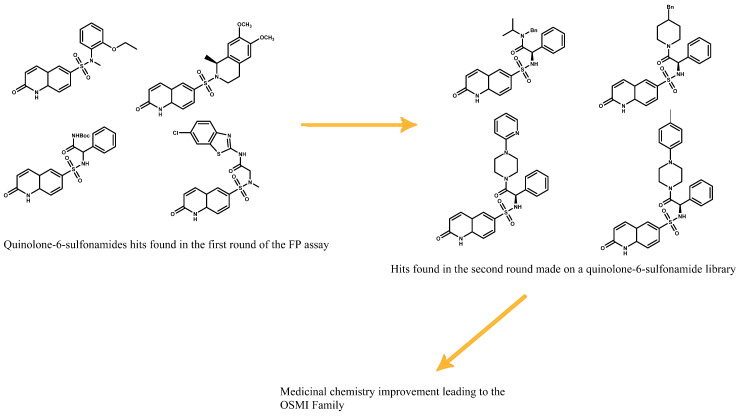
First quinolone-6-sulfonamide hits leading to the development of OSMI family.

**Figure 17 molecules-26-01037-f017:**
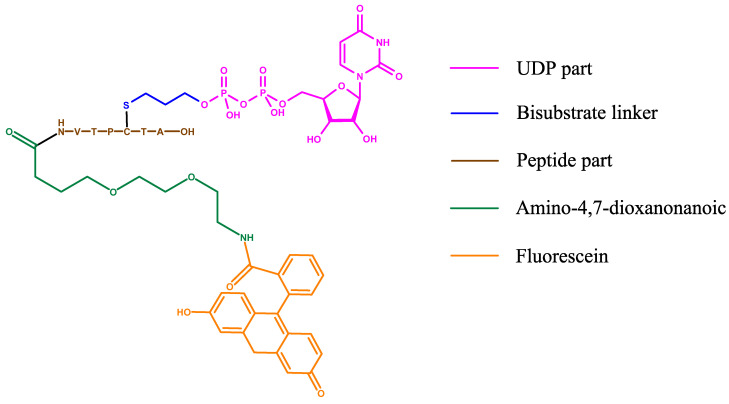
Structure of the bisubstrate probe used in the FP assay by the van Aalten group. The basis of this probe is OGT inhibitor Goblin1 that consists of UDP and a peptide substrate connected by a cysteine-thiol linker. Goblin1 is attached via flexible linker to a fluorescein derivative giving fluorescent properties to the probe.

**Figure 18 molecules-26-01037-f018:**
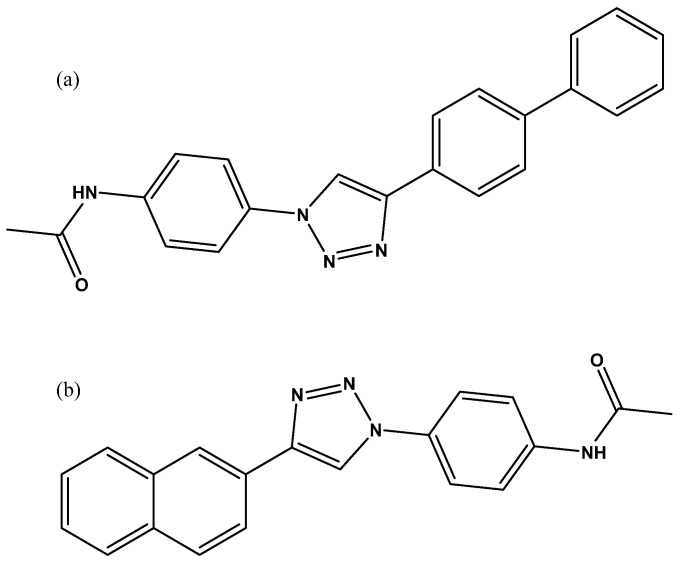
Structure of hits found by the tethering in situ chemistry technique (**a**) APBT; (**b**) APNT.

**Table 1 molecules-26-01037-t001:** Sum up of the pros and cons for all the bioassays mentioned in the review.

Assay Name	Pros	Cons
UDP-Glo assay	-High-throughput screening -Commercially available assay -Standard technique (luminescence)	-Indirect measurement -Involves 3 consecutive enzyme reactions -Possible interference with UDP mimic compounds
[^14^C] UDPGlcNAc	-Direct measurement	-Radioactive reagents -Synthesis of UDP-[1-14C]-GlcNAc
FRET assay	-High-throughput screening	-Use of a second enzymatic step -The protease is not completely selective over the GlcNAcylated peptide
PamStation	-PamChip allows the study of different concentrations of peptide substrate at once -Control of multiple parameters (temperature, reaction time, etc.)	-Not a standard technique -Long time for final readout -Indirect measurement of GlcNAcylation
HPLC	-Standard technique -Direct measurement of GlcNAcylation	-Requires purification of enzymatic reaction -Lengthy purification-to-readout time for each reaction mixture -Use of orthogonal assay to screen hits
FP assay	-One step assay -Very fast and reliable method -High-throughput screening suitability	-Design and synthesis of the probe
Microscale thermophoresis	-Straightforward and easy assay -Quick result	-Specific machine -Not valid for high-throughput screening format (HTS)
Tethering in situ click chemistry	-Allow screening of big libraries -Targeting allosteric sites	-Resynthesis of the hits without the tethering groups -Use of orthogonal assay to remove unactive compounds

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
