# Peer review of "Overview of the Assays to Probe O-Linked β-N-Acetylglucosamine Transferase Binding and Activity"

_molecules, 2021, doi:10.3390/molecules26041037_

Round 1
Reviewer 1 Report
In the manuscript entitled ‘’Probing a ‘small sugar’ transferase – O-linked β-N-acetylglucosamine transferase – with a big assay’, Cyril Balsollier et al. reviewed developed assays and techniques for OGT inhibitor discovery. This reviewer suggests acceptance after revision.
Issues to be addressed:
1) Classic pubs in the O-GlcNAc field and latest articles (especially on evaluation of OGT inhibitors) should be cited, e.g.:
Hart, G. W.; Slawson, C.; Ramirez-Correa, G.; Lagerlof, O. Cross Talk Between O-GlcNAcylation and Phosphorylation: Roles in Signaling, Transcription, and Chronic Disease. Annu. Rev. Biochem. 2011, 80, 825−858.
Hart, G. W. Nutrient Regulation of Signaling and Transcription. J. Biol. Chem. 2019, 294, 2211−2231.
Ma J, Wu C, Hart G.W. 2021. Analytical and biochemical perspectives of protein O-GlcNAcylation. Chem. Rev. https://dx.doi.org/10.1021/acs.chemrev.0c00884
2) Lines 15-16 and lines 62-63: “While some OGA inhibitors are already in phase 3 clinical trials for the treatment of Alzheimer's disease.” & “MK-8719 is currently in phase 3 clinical trials”. Can the authors specify which inhibitors (except MK-8719) are in clinical trials? It would be great if the authors can cite references showing that such inhibitors are indeed in phase 3 clinical trials.
3) Lines 469-471: “The only cellular assay for testing OGT inhibition is a Western blot using anti-O-GlcNAc antibodies to measure the total glycosylation inside the cell.” In this reviewer’s opinion, there might be more than one cellular assay for OGT (e.g., by using unnatural metabolic probes). The authors might want to comment on that a bit?
Author Response
Reviewer 1
In the manuscript entitled ‘’Probing a ‘small sugar’ transferase – O-linked β-N-acetylglucosamine transferase – with a big assay’, Cyril Balsollier et al. reviewed developed assays and techniques for OGT inhibitor discovery. This reviewer suggests acceptance after revision.
Issues to be addressed:
1) Classic pubs in the O-GlcNAc field and latest articles (especially on evaluation of OGT inhibitors) should be cited, e.g.:
- Hart, G. W.; Slawson, C.; Ramirez-Correa, G.; Lagerlof, O. Cross Talk Between O-GlcNAcylation and Phosphorylation: Roles in Signaling, Transcription, and Chronic Disease. Annu. Rev. Biochem. 2011, 80, 825−858.
- Hart, G. W. Nutrient Regulation of Signaling and Transcription. J. Biol. Chem. 2019, 294, 2211−2231.
- Ma J, Wu C, Hart G.W. 2021. Analytical and biochemical perspectives of protein O-GlcNAcylation. Chem. Rev. https://dx.doi.org/10.1021/acs.chemrev.0c00884
As a response to the reviewer's comment, we have cited these 3 references in an appropriate part of the main text.
2) Lines 15-16 and lines 62-63: “While some OGA inhibitors are already in phase 3 clinical trials for the treatment of Alzheimer's disease.” & “MK-8719 is currently in phase 3 clinical trials”. Can the authors specify which inhibitors (except MK-8719) are in clinical trials? It would be great if the authors can cite references showing that such inhibitors are indeed in phase 3 clinical trials.
We have modified the text by describing OGA inhibitors that are already in the phase 2 and 3 clinical trials and added the references to this part:
»In the case of OGA, many potent inhibitors have already been developed to control tauopathy levels in the brain. Two of them are currently in clinical trials for the treatment of Alzheimer's disease and Progressive Supranuclear Palsy (PSP), two diseases associated with tauopathy levels: MK-8719, which has passed preclinical trial tests and, ASN120290, which is for scheduled for phase 2 clinical trials [15,16,17].”
3) Lines 469-471: “The only cellular assay for testing OGT inhibition is a Western blot using anti-O-GlcNAc antibodies to measure the total glycosylation inside the cell.” In this reviewer’s opinion, there might be more than one cellular assay for OGT (e.g., by using unnatural metabolic probes). The authors might want to comment on that a bit?
We agree with the Reviewer 3 and to answer this issue, we have incorporated the text that describes an assay that uses unnatural metabolic probe:
“Another such assay reported by Hu et al. [45] uses the unnatural metabolic probe UDP-6AzGlcNAc with an azide handle, which can be readily detected using click chemistry (attachment of a fluorescent probe with alkyne functionality) and in-gel fluorescence scanning.”
Furthermore, we have cited the next reference: Hu, CW., Worth, M., Fan, D. et al. Electrophilic probes for deciphering substrate recognition by O-GlcNAc transferase. Nat Chem Biol 2017; 13: 1267–1273. doi: 10.1038/nchembio.2494.
Reviewer 2 Report
The review focuses on the development of assays to identify gluc nac transferease inhibitors; molecules that would have much therapeutic interest in neurodegenerative disease
The manuscript reviews these bioassays in a complrehensive and well structure manners; advantages and challenges of the different methods are outlined. It is very well written, covers relevant reports (from the earlier ones to most recent developments) and it will be of interest to scientists working on this area and possibly to the wider medicinal chemistry community.
Minor comments
What is the stereochemistry of the GlucNAc in the protein-alpha or beta-is it relevant for the biological activity?
Indicate in Figure 1- also in text.
The figures and schemes are very clear, however when possible the same settings/styles on the chemical structures should be used throughout the manuscript. Also, in some figures the different structural fragments of the inhibitors/probes have been circled? In different colors; this appears rather messy, is there a way that the authors could convey the same graphical/visual information effectively but in a neater way?
I find the title does not reflect the extent of the information reviewed in the manuscript-it is in fact many more than just a big assay. It does not communicate clearly what the review deals with, and it is indeed a very interesting topic.
Author Response
Reviewer 2
The review focuses on the development of assays to identify gluc nac transferease inhibitors; molecules that would have much therapeutic interest in neurodegenerative disease
The manuscript reviews these bioassays in a comprehensive and well structure manners; advantages and challenges of the different methods are outlined. It is very well written, covers relevant reports (from the earlier ones to most recent developments) and it will be of interest to scientists working on this area and possibly to the wider medicinal chemistry community.
We thank the Reviewer 2 for the compliments and very thorough review.
Minor comments
1) What is the stereochemistry of the GlucNAc in the protein-alpha or beta-is it relevant for the biological activity? Indicate in Figure 1- also in text.
We have modified the Figure 1 accordingly, and have defined the stereochemistry on the anomeric carbon that was lacking in the Figure 1, as well in the figure legend, as follows:
“Figure 1. Dynamic equilibrium of the O-GlcNAcylation process. UDP-GlcNAc with the sugar moiety attached via a-O-glycosidic bond is produced in the hexosamine biosynthetic pathway and subsequently attached to serine/threonine of protein substrates by OGT via O-β-glycosidic bond. It is and removed by OGA, and the GlcNAc formed after the OGA reaction re-enters the hexosamine pathway.”.
We also modified other figures 2,4,7,12 where the stereochemistry was not clear.
2) The figures and schemes are very clear, however when possible the same settings/styles on the chemical structures should be used throughout the manuscript. Also, in some figures the different structural fragments of the inhibitors/probes have been circled? In different colors; this appears rather messy, is there a way that the authors could convey the same graphical/visual information effectively but in a neater way?
As a response to the reviewer comments, we have changed some settings in the figures. We changed also the circles concerning the different chemical probes figures.
3) I find the title does not reflect the extent of the information reviewed in the manuscript-it is in fact many more than just a big assay. It does not communicate clearly what the review deals with, and it is indeed a very interesting topic.
The title has been changed to “Overview of the assays to probe O-linked β-N-acetylglucosamine transferase binding and activity”. We believe it now reflects better the contents of the information covered by the manuscript.
Reviewer 3 Report
The review by Balsollier covers OGT assays used in large scale drug screening experiments. This is a novel review that is informative and valuable. I recommend publishing after some minor grammatical changes. Importantly, the authors need to make sure the letter are capitalized correctly in O-GlcNAcylation. They also should include the O- as well. I recommend defining OGT and OGA in the abstract.
Author Response
Reviewer 3
The review by Balsollier covers OGT assays used in large scale drug screening experiments. This is a novel review that is informative and valuable. I recommend publishing after some minor grammatical changes. Importantly, the authors need to make sure the letter are capitalized correctly in O-GlcNAcylation. They also should include the O- as well. I recommend defining OGT and OGA in the abstract.
We thank the Reviewer 3 for his praise of the manuscript. We have made the changes according to his comments.